# Adult Normative Data for the Adaptation of the Hearing in Noise Test in European French (HINT-5 Min)

**DOI:** 10.3390/healthcare10071306

**Published:** 2022-07-14

**Authors:** Johanna Buisson Savin, Pierre Reynard, Eric Bailly-Masson, Célia Joseph, Charles-Alexandre Joly, Catherine Boiteux, Hung Thai-Van

**Affiliations:** 1Institut de l’Audition, Institut Pasteur, INSERM U1120, 75012 Paris, France; johanna.savin@amplifon.com (J.B.S.); pierre.reynard@chu-lyon.fr (P.R.); charles-alexandre.joly01@chu-lyon.fr (C.-A.J.); 2Amplifon France, 94110 Arcueil, France; eric.baillymasson@amplifon.com (E.B.-M.); celia.1703@hotmail.fr (C.J.); catherine.boiteux@amplifon.com (C.B.); 3Service d’Audiologie et d’Explorations Otoneurologiques, Hospices Civils de Lyon, Hôpital Edouard Herriot, 69003 Lyon, France; 4Faculty of Medicine, University Claude Bernard Lyon 1, 69100 Villeurbanne, France

**Keywords:** speech in noise, speech audiometry, speech intelligibility, adult normative values, European French test

## Abstract

Decreased speech-in-noise (SpIN) understanding is an early marker not only of presbycusis but also of auditory processing disorder. Previous research has shown a strong relationship between hearing disorders and cognitive limitations. It is therefore crucial to allow SpIN testing in subjects who cannot sustain prolonged diagnostic procedures. The objectives of this study were to develop a rapid and reproducible version of the Hearing in Noise Test (HINT-5 min), and to determine its adult normative values in free-field and monaural or binaural headphone conditions. Following an adaptive signal-to-noise ratio (SNR) protocol, the test used a fixed noise level, while the signal level varied to reach the 50% speech reception threshold (SRT50). The speech material consisted of five lists of 20 sentences each, all recorded in European French. The whole semi-automated procedure lasted 5 min and was administered to 83 subjects aged 19 to 49 years with no reported listening difficulties. Fifty-two subjects were retested between 7 and 8 days later. For the binaural free-field condition, the mean SRT50 was −1.0 dB SNR with a standard deviation of 1.3 dB SNR. There was no significant difference between the results obtained at test and retest, nor was there any effect of listening condition, sex, or age on SRT50. The results indicate that the procedure is robust and not affected by any learning phenomenon. The HINT-5 min was found to be both a fast and reliable marker of the ability to understand speech in background noise.

## 1. Introduction

Up to 10% of adults who complain of significant listening difficulties have normal pure-tone thresholds [1]. Currently, hearing assessments in routine audiological practice are mostly performed using pure-tone audiometry accompanied by speech audiometry in quiet. In everyday life, such silent conditions are rare. Hearing discomfort that occurs during social exchanges in a noisy environment may therefore be underestimated [2,3,4]. Indeed, several studies have shown the effect of age on the degradation of intelligibility in noise [5,6,7,8,9,10,11]. Thus, intelligibility assessments are most relevant when performed in the presence of noise, in a so-called more “ecological” manner [2,3,4].

These listening difficulties are attributed to difficulties in sound processing within the central auditory nervous system. Decreased intelligibility in noise is often suggestive not only of incipient presbycusis but also of auditory processing disorder (APD) [12,13,14]. Adults with APD might experience difficulties with auditory processing in daily life, leading to adverse effects such as low self-esteem, anxiety, and depression, with reduced quality of life. Recently, Sardone et al. [15] found a strong association between age-related APD in patients with mild cognitive impairment and dementia, suggesting the involvement of central auditory pathways in neurodegeneration, as pointed out by Illiadou et al. [16,17].

According to Kumar et al., the vast majority of APD adult patients will be dismissed with no further testing beyond speech audiometry in quiet, and with no management recommendations [1]. The main symptoms and anamnesis elements that should lead to the diagnosis of APD have recently been the subject of a European consensus [13]. In children, the value of APD screening and management has also been highlighted [18,19,20]. The diagnostic test battery includes, in addition to speech-in-noise (SpIN) tests, dichotic and binaural integration tests as well as temporal tests [21]. The whole diagnostic procedure is therefore time-consuming [13]. Furthermore, APD is frequently associated with neurodevelopmental or cognitive disorders which are not always compatible with prolonged diagnostic procedures [22,23,24]. It is therefore important to develop supraliminal tests whose administration is compatible with the attentional resources of APD subjects. Failure to assess hearing discomfort in noise may thus prevent the detection of incipient deafness and delay the indication for hearing management and rehabilitation. Furthermore, testing for APD could contribute to the diagnosis of cognitive dysfunction in elderly patients [25] who cannot tolerate prolonged auditory assessment.

Taking into account intelligibility in noise is now recommended or even mandatory in other European countries [26,27,28]. Different SpIN tests are available for adult francophone populations depending on the context in which they are used (hearing screening, audiological diagnosis, or clinical trials) [4,26,29,30]. There is no French-speaking SpIN test for the screening of APD in adults. In children, a SpIN test has recently been described in Greek as an effective screening tool for abnormal auditory processing abilities [31,32]. For diagnostic purposes, recent guidelines from the French Society of Otorhinolaryngology and the French Society of Audiology have recommended the use of adaptive procedures establishing the 50% speech reception threshold in noise (SRT50) to obtain a rapid and standardized measurement [33]. This threshold corresponds to the signal-to-noise ratio (SNR, in dB SNR) necessary for the correct repetition of 50% of the speech material [33,34].

The Hearing in Noise Test (HINT) was one of the first SpIN tests to be based on the calculation of the SRT50 using an adaptive procedure, allowing a fast standardized assessment of intelligibility in noise [35]. It has been validated in many languages around the world [36,37]. A French version has been adapted, but only for adult Canadian francophone populations with normative values obtained from 36 subjects [38,39]. This test is routinely used to evaluate the performance in noise of patients with cochlear implants [40,41,42]. It is also regularly performed when evaluating the effectiveness of a hearing aid, a particular setting, or simply the contribution of binaurality (binaural unmasking property). A multitude of configurations are possible: mono- or binaural, in quiet, in noise, in a free field, or with headphones [26,33,38,39,43]. The HINT lists, currently used in France by hearing care professionals and otorhinolaryngologists to evaluate intelligibility in noise, are derived from the Canadian French version. These lists, although commonly used, have never been standardized for the European francophone population, limiting the use of the test and its validity. Indeed, Vaillancourt et al. recommended that a standard be established for the use of the test in France [38]. As recently pointed out in a systematic review, there is still a need for additional SpIN tests in European French that are both precise and of a reasonable duration [26] to offer to a wide population.

The aim of this study was to validate a European French version of the HINT test in a rapid version (less than 5 min) whose completion time is compatible with the population of interest (presbycusis patients and APD). Adult normative SRT-50 values in free-field, binaural, and monaural conditions were developed.

## 2. Materials and Methods

### 2.1. Subjects

Participants were assessed over a 6-month period in two distinct laboratories. All participants met the following inclusion criteria: born in France, native French speakers, mean tonal hearing thresholds in each ear ranging between 2 and 15 dB with threshold values inferior or equal to 20 dB HL for all frequencies diotically tested per half-octave from 0.25 to 8 kHz. All participants had a normal otoscopic examination and bilateral type A tympanometry. All were free of an ENT history (no middle ear infections; no hearing pathologies such as listening difficulties in noise, tinnitus, hyperacusis, misophonia, phonophobia, disphonia, or diplacusis). An age limit of 50 years was set to limit the potential influence of early-onset presbycusis on the results. None of the participants had been previously exposed to the test.

The hearing thresholds, as well as the intelligibility thresholds in quiet and in noise, were determined in a soundproof booth (decree no. 85590 of 10 June 1985), using an Aurical^®^ audiometer (Natus, Pleasanton, CA, USA) and a TDH-39P headset (Telephonics, Griffon Corporation, New York, NY, USA), or a Delta 100 HR loudspeaker (Siare, RCA Siare Acoustique, Angers, France).

### 2.2. Testing Procedure

After the otoscopy, tympanometry, and tone and speech audiometry in quiet, participants were subjected to the HINT-5 min. The running order of the different conditions tested was randomized (binaural in a free field, binaural in headphones, left ear in headphones, right ear in headphones). To test for the reproducibility of the results, participants were submitted to a HINT-5 min retest 7 to 8 days later, a sufficient time interval to eliminate any learning phenomenon.

### 2.3. Voice and Noise Equipment

The HINT-5 min speech material was extracted from the lists of the Canadian HINT, which are phonetically balanced. Among the 12 original lists, 5 were selected based on their linguistic relevance in European French and then re-recorded by a male speaker for the French Association of Hearing Aid Practitioners [44]. The unit of measurement for scoring the test was the sentence: all the words had to be correctly repeated for the answer to be validated. Only substitutions, omissions, or additions of tool words (prepositions, conjunctions, pronouns, determiners) were allowed. If only one of the lexical words (noun, verb, adjective, adverb) was wrong, the whole sentence was considered to be incorrectly repeated. The noise used was an International Collegium for Rehabilitative Audiology (ICRA)-noise with preserved amplitude modulations. Its spectral image is close to an intermittent noise with an average root mean square (RMS) level equivalent to the first list. ICRA-noise not only represents the spectrum of the speech material but also its modulation [45].

### 2.4. HINT-5 Min Procedure

The HINT-5 min procedure is derived from the HINT version described by Vaillancourt [38]. It consists in an abbreviated version with the advantages of a short completion time, simple speech material (allowing for an increased use of intelligibility in noise testing by clinicians), and an easy and fast interpretation of the test result. Listening difficulties in noise can thus be rapidly and easily detected and measured.

The test can be performed with separate ears using headphones, although the free-field condition is most relevant since speech intelligibility in noise requires binaurality [26]. In the free-field condition, the loudspeaker is placed in front of the patient under test (azimuth 0°), at ear height, and at a distance of 1 m. Both the speech signal and the noise are transmitted from this single loudspeaker.

The noise level is fixed at 60 dB. The speech level is initially set at 65 dB, corresponding to an SNR of +5 dB (thus favorable). It remains fixed for the first 5 sentences corresponding to the training phase which is excluded from the scoring. This phase conditions the patient to listen in noise. From the 6th sentence onwards, each correct or incorrect answer lowers or raises the SNR by 6 dB, respectively, following an adaptive procedure. At the first reversal, a 3 dB step size is then applied. With each correct answer, the speech signal is decreased by 3 dB, while each wrong answer increases the SNR by 3 dB. The extracted SRT50 is calculated after a minimum of eleven sentences tested after the training phase. If the SNR fluctuations are within a 3 dB SNR variation range, the mean SRT50 value is calculated based on the last eight intensity levels presented. The mean SRT50 value is rounded up for decimals greater than or equal to 0.5 (Figure 1a). If the SNR fluctuations exceed an SNR variation range of 3 dB, two mean values are calculated and compared to ensure the stability of the measurement. The first mean value (mean 1), calculated on the last eight intensity levels presented (items 9 to 16), is compared to a second mean value (mean 2), calculated on eight consecutive intensities with a shift of two items (items 7 to 14). If the two rounded mean values are identical, the test stops. If the two mean values have a difference of more than 1 dB SNR, the test continues until mean 1 and mean 2 are identical (Figure 1b).

### 2.5. Statistical Analyses

Statistical analyses were performed using Prism 9 (GraphPad, San Diego, CA, USA). Normal distribution was verified using the D’Agostino–Pearson test. Parametric paired *t*-tests were used in the case of normal distribution. Statistical significance was set at 0.05.

## 3. Results

### 3.1. Participants

Eighty-six subjects were initially recruited for the HINT-5 min normalization study. Three of them were excluded because they had threshold values greater than 20 dB HL. A total of 83 subjects (37 males and 46 females of a comparable mean age) aged between 19 and 49 years (mean (SD) age: 26.3 (7.6) years) were therefore retained for determining the normative values of the test. The mean (SD) tonal values were 8.8 (3.7) dB HL for the right ear and 9.0 (3.9) dB HL for the left ear. The distribution was Gaussian for each ear, and a paired *t*-test showed no significant difference between the values of the two ears. Among all 83 participants, 52 underwent a retest 7–8 days later.

### 3.2. Comparison of SRT50 Scores between Ears

The mean SRT50 scores were obtained for each ear at test and retest. Since the variances were not statistically different and the distribution of the data followed a normal distribution according to the D’Agostino–Pearson test, the parameterized paired *t*-test was used. The results obtained in the right ear condition were not significantly different from those obtained in the left ear condition (*p* = 0.81 and *p* = 0.49 for test and retest, respectively; Table 1).

### 3.3. Comparison of SRT50 Scores between the Binaural Headphone and Free-Field Conditions

The mean SRT50 scores were obtained for both the binaural free-field and binaural headphone conditions at test and retest. The results obtained in the binaural free-field condition were not significantly different from those obtained in the binaural headphone condition (*p* = 0.06 and *p* = 0.84 for test and retest, respectively; Table 2).

### 3.4. Reproducibility of the HINT-5 Min

The reproducibility of the HINT-5 min was tested in a test–retest protocol for all the listening conditions developed. For all four listening conditions tested, there were no significant differences between test and retest (Table 3).

### 3.5. Effect of Age and Sex

The distribution of SRT50 scores obtained in the binaural free-field listening condition was plotted according to age and sex. The unpaired *t*-test found no significant difference between male and female participants (*p* = 0.58; Table 4). Moreover, no correlation was found between the SRT50 score and age (*p* = 0.65, Figure 2).

### 3.6. Normative Values for the HINT-5 Min

Because the test session was more representative of a clinical or audiological hearing assessment than the retest session, only results obtained at test were used for normalization in each listening condition. The mean (SD) SRT50 of the 83 subjects tested in the binaural free-field condition was −1.0 (1.3) dB. Data were normally distributed, with SRT50 values ranging from −3 dB to +2 dB (Figure 3).

## 4. Discussion

The present study established adult normative SRT50 values for the HINT-5 min test administered under different listening conditions: binaural in a free field (−1.0 dB SNR), binaural in headphones (−1.4 dB SNR), and monaural in headphones (−0.8 and −0.8 for the right and the left ear, respectively). No correlations with age or sex were found. For all four listening conditions tested, there were no significant differences between test and retest. The HINT-5 min could therefore be useful for assessing intelligibility in noise in daily life. It could also be very useful for patients with presbycusis—early or not—or APD at the diagnostic stage, or as a follow-up test, especially when fitting a hearing aid.

The observed difference between the normative values of the HINT-5 min (−1.0 dB SNR) and the comparable (0° azimuth) HINT condition described by Vaillancourt (−3.0 dB SNR) [29,30] could be explained by several discrepancies between the two tests. Indeed, the vocal material was re-recorded using a European French native speaker. Moreover, a different noise was used. Rather than the LTASS-noise—a filtered white noise resulting in a long-range spectrum of speech noise—used in Vaillancourt’s procedure, the HINT-5 min uses an ICRA-noise which represents not only the spectrum but also the modulation of the speech material, where LTASS only reproduces the spectrum. The duration of the test is very short (less than 2 min per condition), in order to minimize the impact of the subject’s attentional fluctuation and to evaluate the real intelligibility in a noisy environment. The listening conditions must therefore be more difficult, hence the use of a more masking sound avoiding ceiling effects and the mixing of the two sources from the same loudspeaker, at azimuth 0°. Mixing the signal with the noise in the same loudspeaker makes the test more difficult compared to the experimental condition in which the sources are separated [29]. However, it allows the test conditions to be easily reproduced in different laboratories. This criterion of reproducibility is crucial in the context of multicenter studies in different laboratories, in France or across French-speaking countries. In addition, this condition could facilitate the use of the HINT-5 min for the purpose of evaluating the benefit of hearing equipment by focusing on the quality of the hearing aid setting and not on the directional skills of the microphones. It therefore makes it possible to actually test the quality of the hearing aid settings as opposed to the previous use of the tests in hearing aid evaluation [45,46,47,48,49,50]. In order to avoid fluctuations in the hearing aid’s signal processing algorithms with each change in sound level that occurs, fixed noise is preferred. Therefore, the evaluated criterion is not the hearing aid’s ability to adapt to the variation in the signal-to-noise ratio, but rather the signal emergence due to the personalized setting. The HINT-5 min procedure, consequently, involves varying the signal intensity level, leaving the noise level fixed.

The test can be performed in a free field (using a single speaker) or with headphones in binaural mode, depending on the equipment available in the office. Although headphones can be used to test each ear separately, one has to keep in mind that the most ecological procedure, as well as the best way to take binaural unmasking into account, is to perform binaural testing in a free field [33]. Further studies are needed to evaluate the HINT-5 min in free-field conditions with two loudspeakers to deliver the speech signal and noise separately. This is mandatory to reproduce physiological interaural differences in intensity and time, and to allow the assessment or comparison of gain with hearing aids [33].

In practical terms, the main advantage of the HINT-5 min is its duration. Its semi-automated adaptive performance makes it a short test, easily integrated into a complete hearing assessment. A binaural test in a free field or with headphones will take less than 2 min (with instructions). For a more complete assessment aiming at evaluating intelligibility in binaural free-field conditions as well as the performance of each ear in headphones, the test will take less than 5 min. This test could complement the examinations performed in the diagnostic phase of APD patients alongside less sentence-based tests to limit mental substitution. This is also particularly relevant in the elderly, a population largely represented in consultation for presbycusis assessment. The potentially limited attentional resources require the use of rapid tests. Aging subjects cannot sustain prolonged testing and are prone to fatigue. It could be interesting to have noise speech audiometry tools with a fast procedure suitable for routine use in the office. Such a test could improve the detection of hearing difficulties in noise in the many patients who present with a complaint of comprehension in noise, without necessarily having a loss in pure-tone audiometry. We believe that this test could improve the early detection of early presbycusis and APD. There is an increasing probability of “cognitive decline” with the degree of deafness and with age. The cognitive decline probability reaches 65% for the 55–69 age group and increases to 79% for the 70+ age group [51]. In the range of available tests in noise, this fast and reliable test (test–retest reproducibility) also seems more appropriate for patients exhibiting cognitive disorders, and should be superior to a longer test, since it is likely to be less affected by the cognitive state of the tested subject.

It is noteworthy that the test–retest comparison did not show any significant variability between the two test sessions for subjects aged between 19 and 49 years, and for whom the level of concentration, not measured in the present protocol, was inevitably different. The HINT-5 min procedure is reproducible and is not limited by any learning phenomenon. It could therefore be used repeatedly for hearing assessments, the fitting of hearing aids, and follow-ups. In the case of a patient wearing a hearing aid, the use of the five training lists administered before the actual test procedure allows the prosthesis to detect the “speech + noise” condition. However, the test does not free itself from the mental substitution phenomenon and therefore cannot be used to determine phonemic confusions in noise. Like all tests involving mental support, it is sensitive to the subject’s capacity to focus on the task.

## 5. Conclusions

The present study established the HINT-5 min normative data for speech-in-noise assessment in binaural free-field, binaural headphone, and monaural right and left ear listening conditions for European French-speaking adults under 50 years of age. No significant difference in mean SRT50 values was observed between the free-field and headphone binaural listening conditions as well as in the test–retest condition. The test is robust and is not limited by any learning phenomenon. There was also no correlation of SRT50 with the sex or age of the patient. Furthermore, as the duration of this test does not exceed 5 min, it could be adapted to the diagnostic phase, particularly in patients with reduced cognitive load. The HINT-5 min is a good marker of overall ability to understand speech in a noisy environment in everyday life and could therefore be widely used to help presbycusis and APD diagnosis, as well as hearing aid settings. Nonetheless, further studies are needed on an APD population in order to proof test this short version of the HINT test.

## Figures and Tables

**Figure 1 healthcare-10-01306-f001:**
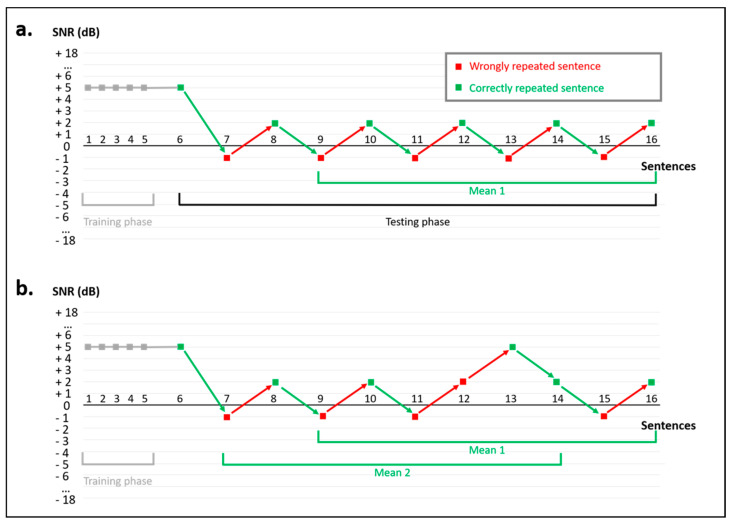
Adaptive procedure for the HINT-5 min. The calculation of the speech reception threshold depends on the magnitude of the signal-to-noise ratio (SNR) changes, which can either be within a range of 3 dB (**a**) or exceed this range (**b**).

**Figure 2 healthcare-10-01306-f002:**
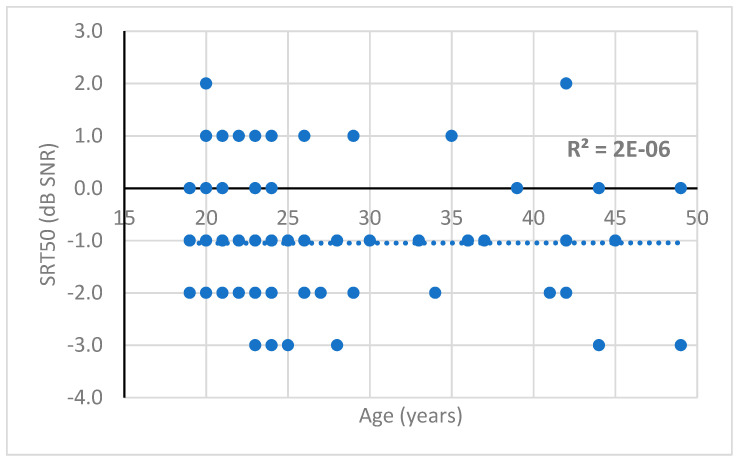
Binaural free-field SRT50 scores obtained during the HINT-5 min according to age. Regression curve with the correlation factor r^2^ = 2 × 10 ^−6^.

**Figure 3 healthcare-10-01306-f003:**
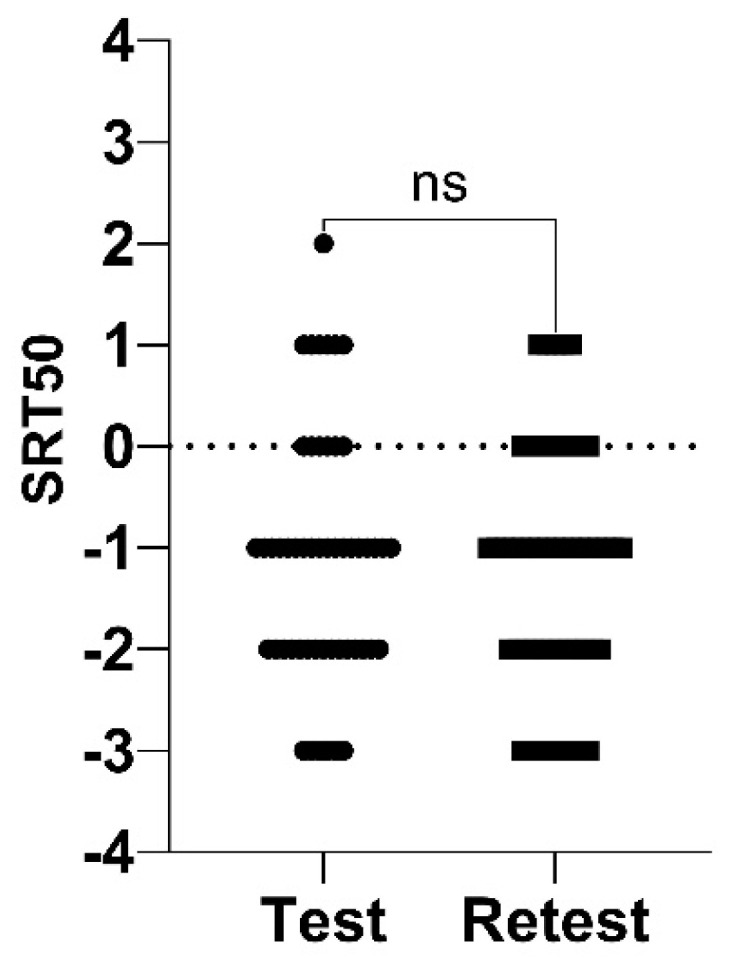
Distribution of SRT50 values obtained during the HINT-5 min at test and retest.

**Table 1 healthcare-10-01306-t001:** Mean SRT50 values for the right and left ears in headphone listening conditions.

	Right Ear	Left Ear	Paired *t*-Test
Test (*n* = 83), mean (SD)	−0.8 (±1.3)	−0.8 (±1.4)	*p* = 0.81 (ns)
Retest (*n* = 52), mean (SD)	−0.9 (±1.5)	−1.0 (±1.3)	*p* = 0.49 (ns)

Values are expressed in dB SNR. A paired *t*-test was performed on the results obtained from the right and left ears for both the test and retest sessions. SD: standard deviation; ns: non-significant.

**Table 2 healthcare-10-01306-t002:** Mean SRT50 scores for the binaural free-field and headphone listening conditions.

	Binaural Free Field	Binaural Headphones	Paired *t*-Test
Test (*n* = 83), mean (SD)	−1.0 (±1.3)	−1.4 (±1.4)	*p* = 0.06 (ns)
Retest (*n* = 52), mean (SD)	−1.2 (±1.2)	−1.3 (±1.3)	*p* = 0.84 (ns)

Values are expressed in dB SNR. A paired *t*-test was performed on the results obtained from the free-field and headphone conditions for both the test and retest sessions. SD: standard deviation; ns: non-significant.

**Table 3 healthcare-10-01306-t003:** Mean SRT50 scores obtained during the test and retest sessions for the different listening conditions developed in the HINT-5 min validation protocol.

	Test (*n* = 52)	Retest (*n* = 52)	Paired *t*-Test
Binaural free field, mean (SD)	−1.1 (±1.2)	−1.2 (±1.2)	*p* = 0.65 (ns)
Binaural headphones, mean (SD)	−1.6 (±1.5)	−1.3 (±1.3)	*p* = 0.14 (ns)
Right ear headphones, mean (SD)	−0.6 (±1.5)	−0.9 (±1.5)	*p* = 0.42 (ns)
Left ear headphones, mean (SD)	−0.7 (±1.4)	−1.0 (±1.3)	*p* = 0.21 (ns)

Values are expressed in dB SNR. A paired *t*-test was performed on the results obtained during the test and retest sessions for each listening condition. SD: standard deviation; ns: non-significant.

**Table 4 healthcare-10-01306-t004:** Mean SRT50 scores for both male and female populations in the binaural free-field listening condition.

	Female (*n* = 46)	Male (*n* = 37)	*t*-Test
Mean (SD)	−1.0 (±1.2)	−1.1 (±1.4)	*p* = 0.58 (ns)

Values are expressed in dB SNR. Unpaired *t*-test was applied between male and female participants. SD: standard deviation; ns: non-significant.

## Data Availability

The data presented in this study are available on request from the corresponding author. The data are not publicly available due to ethical, legal, or privacy issues.

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
