# Peer review of "Adult Normative Data for the Adaptation of the Hearing in Noise Test in European French (HINT-5 Min)"

_healthcare, 2022, doi:10.3390/healthcare10071306_

Round 1

Reviewer 1 Report

The paper argument is interesting and the analysis shown are correct. By the way, there are some points to be improved that would seriously improve the communicative aspects of the work.

The abstract does not provide a good introduction to the potential importance of the paper. Improvements can be applied by specifying why the tests are important and use. Provide then a couple lines of background information and why the study improves the background.

Typos occur in the text. Please follow journals rules and improve a little the English writing style.

References should be inserted in square brackets.

The international scientific community is not interested in the French method, nor its validation. That means that the focus of the paper should be shifted toward other aspects more interesting for an international audience. This does not means the argument should be changes, it is just the idea on “how selling it” that should be reworked toward making the readers more attracted by the paper.

Conclusions are too shorts.

Reference section can be a bit improved reporting more background in the intro.

Reviewer 2 Report

83 normal hearing adults 19-49 years (pure tone audiometry, tympanometry, no subjective hearing problems) were tested for French sentence recognition in 60db babble noise. The SRT50 method intends to identify which speech signal level enables 50% correct recognition of the words. Using four conditions (free field, double earphone and each ear separately) stable values around -1 db compared to the noise were found. 52 subjects were tested about 8 days later with no significant difference in results. The conclusion is that the quick 5 min test has so well defined normative values that it should be able to identify subjects with disorders of auditory processing, or simply age related changes.

My comments on this well written, easy to follow, logical paper is that it is not clear how the proposed correct/incorrect results in hearing the tested sentences/words correspond to a 50% success rate – the paper just describes the increase and decrease of speech level until some stability is reached. Please expand to the readers.

Moreover, why do values differ to the Canadian French version : -3db in Canada compared to -1db in this French test. Different noise methods are mentioned, but as the differences are highly significant this raises questions on the methodology.

Finally, if the usefulness to diagnose auditory processing disorders is the aim of the work, do the authors have any preliminary data to support that the method could be useful?
